# Multimodal-aware Multi-intention Learning for Recommendation

## ABSTRACT

Whether it is an e-commerce platform or a short video platform, the effective use of multi-modal data plays an important role in the recommendation system. More and more researchers are exploring how to effectively use multimodal signals to entice more users to buy goods or watch short videos. Some studies have added multimodal features as side information to the model and achieved certain results. In practice, the purchase behavior of users mainly depends on some subjective intentions of users. However, it is difficult for neural networks to effectively process noise information and extract high-level intention information. To investigate the benefits of latent intentions and leverage them effectively for recommendation, we propose a Multimodal-aware Multi-intention Learning method for recommendation (MMIL). Specifically, we establish the relationship between intention and recommendation objective based on probability formula, and propose a multi-intention recommendation optimization objective which can avoid intention overfitting. We then construct an intent representation learner to learn accurate multiple intent representations. Further, considering the close relationship between user intent and multimodal signals, we introduce modal attention mechanisms to learn modal perceived intent representations. In addition, we design a multi-intention comparison module to assist the learning of multiple intention representations. On three real-world data sets, the proposed MMIL method outperforms other advanced methods. The effectiveness of intention modeling and intention contrast module is verified by comprehensive experiments.

## CCS CONCEPTS

• **Information systems** → **Recommender systems**.

## KEYWORDS

Multiple Intentions, Multimodal Recommendation, Contrastive Learning

**ACM Reference Format:**
Anonymous Author(s). 2024. Multimodal-aware Multi-intention Learning for Recommendation. In *MM '24: ACM International Conference on Multimedia, October 28 - November 01, 2024, Melbourne, Australia.* ACM, New York, NY, USA, 9 pages. https://doi.org/10.1145/nnnnnnn.nnnnnnn

**Unpublished working draft. Not for distribution.**

## 1 INTRODUCTION

Benefiting from the improvement of data transmission and storage performance, more and more rich modal data is used in various applications [6]. E-commerce platform is one of the most full use of multi-modal information scene, providing text description of goods, picture style and corresponding video explanation [31]. Short video programs are one of the most popular applications in recent years, which are natively composed of multimodal data [34]. Each video has detailed introduction, rich visual signals, and engaging audio information. Whether it is an e-commerce platform or a short video platform, the effective use of multi-modal data plays a vital role in the recommendation system. More and more researchers begin to explore how to effectively use multi-modal signals to improve the accuracy of the recommendation system and attract more users to buy goods or watch short videos [1, 11, 14].

In recent years, many researches have added multi-modal features as side-info information to models and achieved certain results [4, 7]. They encode images using visual encoders such as ResNet [13] and ViT [8], and model user visual preferences using attentional mechanisms. Hierarchical CNN are used to conduct semantic characterization of product reviews and titles, so as to extract users' personalized semantic preferences [4]. Then visual representation and text representation are used as side-info features to carry out feature interaction with users and item attributes. Since the image representation supplements the visual attraction signal of the product to the user, and the text representation supplements the semantic attraction signal of the product, the use of these modal data improves the effect of the recommendation system [3, 7]. In addition, there are some work using graph neural networks to effectively capture modal high order connectivity signals [41, 45]. For example, MMGCN [34] performs graph convolution operations on each bipartite graph to learn representation vectors.

In the actual scenario, users often have their own intentions when buying goods [5]. For example, one user is a 24-year-old female college student. When she goes shopping on the e-commerce platform in summer, she tends to buy the latest fashion skirts and clothes with beautiful colors. This reflects the user's purchase intention of the required style and color before shopping. Considering that intention is a preference expression at a higher level than the original feature, it can better reflect the characteristics of user's personalized selection. Therefore, this paper focuses on how to construct accurate representations of intent and provide personalized recommendations based on intent. In addition, whether it is image, text or audio signals, they often contain a lot of noise information. Many of the previous work [4, 37] directly use the representation of high noise for model learning, which is easy to damage the recommendation effect. Therefore, it is necessary to extract a high-level representation of intention based on multimodal data.

However, simply extracting intent representations from data is unfriendly because neural networks cannot effectively process

noisy information and extract high-level semantic representations. Moreover, there are potential problems with intent-based recommendations. In particular, models may overfit to some intentions since there are usually a limited number of intentions and some intentions may be highly weighted. For example, the model easily learns a relationship between *A: This user is a mom* and *B: this user tends to buy kitchenware* according to high frequency similar behavior. However, it is also possible for a mum to buy fashionable new dresses, making the above correlation untenable in other intentions. In this work, we call such issue as intention overfitting, indicating that recommendation models may overfit to some intentions with unreliable correlations, and thus give wrong predictions on the other intentions.

Considering the problems mentioned above, we propose a new multi-intent modeling scheme. We propose a Multimodal-aware Multi-intention Learning (MMIL) method for recommendation. Specifically, we first propose a multi-intent recommendation framework that can avoid intent overfitting, based on the probability distribution relationship among predict targets and intents. We transform the intentional-recommendation objective optimization into the lower bound optimization according to the Jensen Inequality. We then construct an intent representation learner to learn accurate multiple intent representations. Further, considering the close relationship between user intent and multimodal signals, we propose a modal information perceptron to learn modal - aware intent representation. In addition, we design a multi-interest contrastive module to assist the learning of multi-interest representation. Finally, we combine the above optimization objectives as a comprehensive loss function for model training.

The main contributions of this paper can be summarized as follows:

- We propose a multi-intent recommendation scheme that avoids intent overfitting, and transform the optimization objective into a model-friendly optimization lower bound by Jensen's inequality.
- We propose a multimodal-aware multi-intention learning method (MMIL) for recommendation. MMIL can not only construct multi-intention representation of users, but also learn intention information of modal attention by using modal perceptron.
- We have carried out rich experiments on three real-world public data sets, and the experimental results show that our proposed MMIL method outperforms the state-of-the-art methods. Further experimental analysis also verifies the effectiveness of our proposed multi-intention recommendation module and intention contrastive module.

## 2 RELATED WORK

### 2.1 Multimodal Recommendation

The method based on collaborative filtering has been widely used in early recommendation systems and has achieved certain results [20, 28]. However, due to the impact of data sparsity and other factors, collaborative filtering model has been unable to meet the needs of large-scale recommendation systems. In recent years, with the increasing abundance of multimodal data, many researchers began to explore the use of multimodal signals, so as to alleviate the problem of data sparsity in recommendation systems [11, 38, 41]. Multimodal recommendation systems are widely used in e-commerce and short video platforms.

Many multimodal recommendation models mainly learn multi-modal information of items as side information [4, 7]. The effect of the model is improved by the interaction of multimodal features and attribute features with high order features. For example, VBPR [15] extracts visual features from product images and incorporates visual signals into matrix factorization models. DVBPR [19] improves recommendation performance by jointly training the image representation and recommendation system from the pixel level. ACF [3] proposes an attention mechanism based on the item layer and component layer to deal with recommendation tasks in the multimedia domain. VECF [4] also models the user's attention perception information for different areas of the image and review. More recently, fine-grained modeling methods have improved multimodal recommendations [10]. SCAHN [23] designs a semantic structure-enhanced contrastive adversarial hash network to enhance model representation learning.

Recently, many research methods using graph neural network have achieved good results. MMGCN [34] constructs a user-item dichotomous graph on each mode and enriches the representation of each node with the topology and characteristics of its adjacent nodes. GRCN [33] adaptively adjusts the structure of the interaction graph according to the training state of the model, and applies the graph convolution layer to the refined graph to extract the user's preferred information signal. HUIGN [32] presents user intentions in a hierarchical graph structure from fine to coarse-grained. It achieves multiple levels of user intent by recursively performing intra-level and inter-level aggregation operations.

### 2.2 Contrastive Learning for Recommendation

Self-supervised learning has been widely used in computer vision, recommendation system and other fields in recent years [18, 25, 42]. As an important branch of self-supervised learning, contrast learning can obtain robust and discriminant feature representations by constructing enhanced positive and negative sample pairs. For visual signals, a lot of work has been done to construct enhanced samples by discarding, flipping, masking, etc., and to improve model performance by contrast loss optimization [9, 12].

Many sub-direction studies in the recommendation system have achieved good results through comparative learning [22, 39, 46]. CLS4Rec [36] designs three data enhancement modes of item crop, item mask and item reorder, and improves the sequence recommendation effect by constructing contrastive objectives. ICL [5] defines user intention by means of clustering, and improves recommendation effect based on intention contrastive learning. MHCN [40] integrates self-supervised learning into the training of hypergraph convolutional networks to obtain the connectivity information that maximizes hierarchical mutual information. SLMRec [29] designs three data augmentation methods such as feature dropout and feature masking. SLMRec enables multimedia recommendation models to better establish modal associations and learn stronger multimodal representations.

Recently, many methods combining graph neural network and contrast learning have been widely studied [34, 41]. SGL [35] explores the application of SSL on user-item biparts, and applies self-discrimination to learn more robust node representation, thus assisting the recommendation model to improve its effectiveness. GPT-GNN [17] designs a self-supervised property graph generation task for GNN pre-training to model both structure and properties of the graph, and then uses the pre-training GNN of the input graph as model initialization for different downstream tasks. CFAA [24] proposes a self-supervised contrastive multimodal alignment task to make full use of cross-modal text and visual information. MMGCL [39] builds data augmentation based on modal edge discarding and modal mask, and then uses multiple views of users and projects to construct contrastive learning objectives.

## 3 PRELIMINARIES

We assume that there are a set of $M$ users $U$ and a set of $N$ items $I$. We denote a user-item interaction matrix $Y \in R^{N \times M}$ to represent the interaction relations, where $y_{ui} = 1$ indicates user $u \in U$ has positive feedback with item $i \in I$ before, otherwise $y_{ui} = 0$. Besides, multimodal features are offered as content information of items. In this paper, we consider visual, textual and acoustic modalities. We denote the multimodal features of items as $z^m$, where $m \in M = \{v, t, a\}$. $v$ denotes visual features, $t$ denotes textual features and $a$ denotes acoustic features. The objective of model learning is to learn user and item embeddings based on historical interactions and multimodal features, where the internal product between user embeddings and item embeddings represents the likelihood of users selecting projects. Finally, we recommend the top-$n$ items to users.

## 4 THE PROPOSED MMIL MODEL

In this section, we give a detailed introduction to the proposed MMIL. The MMIL consists of four main parts. Firstly, we establish the relation between intention and recommendation objective based on probability formula, and propose a multi-intention recommendation optimization objective which can avoid intention overfitting. Secondly, we construct an intention representation learner to learn accurate multiple intention representations. Thirdly, considering the close relationship between user intent and multimodal signals, we introduce modal attention mechanisms to learn modal perceived intent representations. Fourthly, we also design a multi-intention comparison module to assist the learning of multi-intention representation. Finally, we combine several optimization objectives for model training.

### 4.1 Multi-intention Recommendation Framework

Users have multiple intentions when they buy goods or watch short videos. To avoid the model being overly influenced by some high-frequency intent shown in Fig. 1., the intuitive idea is to intervene with $Z$ so that $Z$ can avoid being influenced by $I \to Z$. For example, we can mitigate the confounding bias by random sampling. However, due to the limited resources in practice, the efficiency and stability of this method are relatively low.

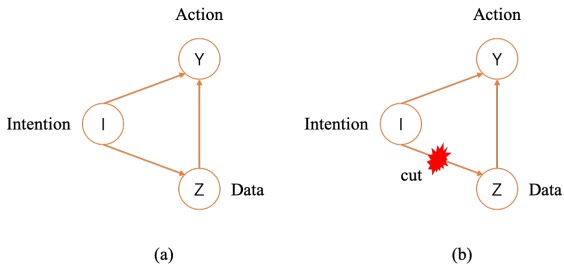

(a)      (b)

**Figure 1: Causation based on intent recommendation. (a) To learn intentions without considering the confounding influence between I and Z. (b) To learn intentions from data and cuts off the confounding effect caused by I->Z.**

Therefore, we consider cutting off the connection between data Z and intent I. Specifically, we define $P(Y|I(Z))$ as objective function, then $P(Y|I(Z))$ can be represented as:

$$
\begin{aligned}
P(Y|I(Z)) &= \sum_i P(Y|I(Z), i) P(i|I(Z)) \\
&= \sum_i P(Y|Z, i) P(i|I(Z)) \\
&= \sum_i P(Y|Z, i) P(i)
\end{aligned}
\tag{1}
$$

where $i$ indicates the intention of the user. The above transformation is mainly based on conditional probability formula transformation. In particular, since we cut off the connection of $I$ and $Z$, which can be thought of $I$ and $Z$ is independent of each other, thus satisfy $P(i|I(Z)) = P(i)$.

However, the actual intentions cannot be observed. The difficulty is that we need to build semantically accurate representations based on data to approximate representations of intents. Therefore, we introduce the distribution of $Q(I|X)$ to approximate the prior $P(I)$, so as to solve the problem above. Specifically, we can deduce the following based on Jensen's inequality:

$$
\begin{aligned}
log \sum_i P(Y|Z, i) P(i) &= log \sum_i Q(i|Z) P(Y|Z, i) \frac{P(i)}{Q(i|Z)} \\
&\geq \sum_i Q(i|Z) log P(Y|Z, i) \frac{P(i)}{Q(i|Z)}
\end{aligned}
\tag{2}
$$

Therefore, we can obtain the optimization lower bound as:

$$
E_{Q(I|Z)} log P(Y|Z, i) - D_{KL}[Q(I|Z)||P(I)]
\tag{3}
$$

### 4.2 Intent Representation Learner

Due to the distribution of $Q(I|Z)$ is unknown, we need to design an intention representation module to approximate distribution of $Q(I|Z)$. Simplified, we can learn $Q(I|Z)$ by clustering method. The core idea is to calculate the distance between each sample and each intent vector. We believe that a user often has multiple intentions in the process of buying and watching, but the importance of different intentions is different. Therefore, we construct a soft cluster to describe the probability that a sample belongs to each intent. In this case, the intent module can use soft distance to learn distribution $Q(I|Z)$, which is defined as follows:

$$Q_s(i|z) = P(i|x) = \frac{e^{-g(f(z),i)}}{\sum_{i'} e^{-g(f(z),i')}} \quad (4)$$

where $i$ represents the learnable intent representation, $g$ represents the distance function, and $f$ represents the metric space mapping function. The main reason we introduce the mapping function $f$ is to consider that the weights of different feature dimensions should be different. This idea comes from design in related studies [40].

Further we set $s_j^k$ equal to $Q(I|Z)$, which is the probability that the $j$-th sample contains the $k$-th intent. Our learning objective is to minimize the distance expectation from each sample to the intent vector, defined as follows:

$$L_d = \frac{1}{N} \sum_{i=1}^{K} \sum_{j=1}^{N} s_j^k g(f(z_j), i_k) \quad (5)$$

In addition, we believe that intents satisfy the assumption of conditional independence. In order to avoid homogeneity among multiple intents, which will reduce the diversity of intents, we introduce the intention constraint function to improve the orthogonality among the representations of intents. The constraint loss function is defined as follows:

$$L_c = \sum_{m=1}^{K} \sum_{n=n+1}^{K} (i_m * i_n) \quad (6)$$

### 4.3 Modal-specific Perceptron

In multimodal condition, the user's intention is greatly affected by visual and auditory signals. We propose the modal awareness mechanism and substitute it into Formula 2. We consider both the modal independent attribute modality and the modal dependent visual and text modalities. Specifically, we define $P(Y|I, Z)$ as follows:

$$P(Y|I, Z) = \sum_{m \in M = \{v,t,a\}} P(m|I, Z)P(Y|m, I, Z) \quad (7)$$

where $M$ represents different modal features, including visual feature $v$, text feature $t$ and attribute feature $c$. Then we can rewrite Formula 3 as follows:

$$E_{Q(I|Z)} \log \sum_m P(m|I, Z)P(Y|m, I, Z) - D_{KL}[Q(I|Z)||P(I)] \quad (8)$$

where $m \in M = \{v, t, c\}$ denotes the modal flag. To solve the above formula, we introduce an intention inference module to calculate each modal intention probability of a user. Considering the power of multi-head attention mechanisms to capture context-important semantic information, we use transformer [30] block to learn intent-dependent modal probability. The definition is as follows:

$$r = f_r(Multi - head - attention([e_c, e_v, e_t])) \quad (9)$$

where $r \in R^{1 \times 3}$ represents the probability distribution of intent related to the modality, approximately $P(m|I, Z)$. $f_r$ represents a mapping function that converts representations into the weight vector, implemented with a fully connected neural network with softmax. $e_m$ represents the semantic representation of the $m$-th modality, obtained by modal encoder $g_m$.

Further, we consider using modal information to perceive intent representations and establish the relationship between modal semantics and intents. We take the intention representation matrix $I$ as query and the modal representation $E^m$ as key and value, and use an attention-perception module to extract the modal intention information as follows:

$$V^m = softmax(\frac{(I_m W_q)(e_m W_k)^T}{\sqrt{d_k}})(e_m W_v) \quad (10)$$

where $V^m \in R^{K \times d}$ denotes the modal-aware intention representation matrix. $W_q$, $W_k$ and $W_v$ represent parameter matrices. Therefore, we can obtain the representation of the $j$-th sample $z_j$ under modal perception of the $k$-th intention as follows:

$$h_j^k = \sum_{m=1}^{3} r_m V_k^m \quad (11)$$

where $h_j^k \in R^{1 \times d}$ represents modal perception intention.

### 4.4 Multi-intention Contrastive Regularization

Due to the problem of sparse data in large-scale recommendation systems, the feature representation cannot be adequately trained. Therefore, in order to further improve the accuracy of the intention representation constructed by us, we use the self-supervised contrast learning method to improve the model training effect. In particular, given the historical interaction sequence $s_u$ of user $u_j$, we construct the enhanced sample by random sampling. Two subsequences $s'_u, s''_u$ are obtained by random sampling of the interaction sequence twice with sampling probability $\mu$. Then, based on two sequence features, multi-modal features and user attribute features, we use the above multi-intent extractor to construct multi-intent representations as follows:

$$\begin{aligned} H' &= Multi - Intent - Extractor(\{s'_u, e'_m, e_u\}) \\ H'' &= Multi - Intent - Extractor(\{s''_u, e''_m, e_u\}) \end{aligned} \quad (12)$$

Then we construct comparative learning modules based on two kinds of multi-intention representations. For user $u_j$, we construct a positive sample pair as $< h'_j, h''_j >$ and $2K - 2$ negative sample pairs composed of other different intentions. It is common practice to construct a comparative loss by taking the intent of other users in the same batch as a negative sample. But there is no guarantee that intentions will be different from one user to another. In addition, considering that K may be a large number, constructing negative samples in batch dimension will increase the computational complexity of the model. Therefore, we construct multi-intent contrast losses only in the single user dimension. Let the set formed by $2K - 2$ negative sample pairs be $S^-$, the multi-intention contrast loss is defined as follows:

$$L_{cl} = -\log \frac{exp(sim(h_j^{k'}, h_j^{k''})/\tau)}{exp(sim(h_j^{k'}, h_j^{k''})/\tau) + \sum_{s \in S^-} exp(sim(h_j^{k'}, s)/\tau)}$$

$$- \log \frac{exp(sim(h_j^{k'}, h_j^{k''})/\tau)}{exp(sim(h_j^{k'}, h_j^{k''})/\tau) + \sum_{s \in S^-} exp(sim(h_j^{k''}, s)/\tau)} \quad (13)$$

By optimizing multi-intent contrast loss on augmentation samples to help model training, users' multi-intent learning is no longer sensitive to some specific positive samples. By reducing the interference of noise interaction, the model can be more effective in training and alleviate the problem of data sparsity.

## 4.5 Prediction

The final prediction results are learned based on the captured sample multi-intention information. Based on the intention probability $s_j^k$ and the intention representation $h_j^k$ of modal perception, the final prediction result is:

$$\hat{p}_j = f_p(\sum_{k=1}^{K} s_j^k h_j^k) \tag{14}$$

where $\hat{p}_j \in [0, 1]$ denotes the prediction result. $f_p$ represents the encoder, directly using a three-layer MLP with softmax function. Therefore, based on the predict value $\hat{p}_j$ and real label $y$, we construct the cross entropy loss function as follows:

$$L_p = -\frac{1}{N} \sum_{j=1}^{N} (y_j log\hat{p}_j + (1 - y_j)log(1 - \hat{p}_j)) \tag{15}$$

## 4.6 Training and Optimization

The ultimate learning goal of the model is to maximize the goal in Formula 6. We can solve the first item by optimizing the final estimated loss $L_p$. For the KL loss of the second item, we can redefine it as follows:

$$\begin{aligned} D_{KL} &= \frac{1}{N} \sum_{k=1}^{K} \sum_{j=1}^{N} s_j^k log(\frac{s_j^k}{P(I)}) \\ &= \frac{1}{N} \sum_{k=1}^{K} \sum_{j=1}^{N} s_j^k (logs_j^k + logK) \end{aligned} \tag{16}$$

where we assume that the intention distribution follows a uniform uniform distribution [xx]. Further, we can get the KL loss function as:

$$L_{kl} = \frac{1}{N} \sum_{k=1}^{K} \sum_{j=1}^{N} s_j^k slogs_j^k \tag{17}$$

Finally, multiple optimization objectives mentioned above are integrated as the final learning objectives of the model as follows:

$$L = L_p + L_{kl} + \mu L_d + \theta L_c + \gamma L_{cl} \tag{18}$$

where $\mu$, $\theta$ and $\gamma$ represent hyper-parameters used to control the important weight of different loss functions. By optimizing the fusion losses mentioned above, the model can learn accurate multi-intention representation and make effective recommendations.

## 5 EXPERIMENTS

In this section, we conduct extensive experiments to answer the following questions:

- **RQ1** Compared with the state-of-the-art multimedia recommendation frameworks, how does our MMIL model perform?
- **RQ2** What is the impact of the different modules in the MMIL on the model effect?

**Table 1: Statistics of the three datasets.**

| Dataset | User | Item | Interactions | Density |
|---------|------|------|--------------|---------|
| Clothing | 39,387 | 23,033 | 237,488 | 0.026% |
| Sports | 35,598 | 18,357 | 256,308 | 0.039% |
| Baby | 19,445 | 7,050 | 139,110 | 0.101% |

- **RQ3** How sensitive is our model under the perturbation of several key hyper-parameters?

## 5.1 Experimental Settings

*5.1.1 Datasets.* We select three datasets[1] from the Amazon product dataset[21], including Clothing, Shoes and Jewelry, Sports and Outdoors, and Baby. These datasets include rich meta information about users and items, such as item descriptions and item images. The visual features are published and represented as 4,096-dimensional embeddings. Following [2], we concatenate the title, descriptions, categories, and brand of each item to extract textual features. Text feature embedding is generated by Sentence-Bert [27]. The statistical results of the three datasets after preprocessing are shown in Table 1.

*5.1.2 Baselines.* To evaluate the performance, we compared the proposed MMIL model with the following baselines:

- **VBPR** [15] integrates visual features as auxiliary signals into biased MF method to improve the accuracy of item representation, especially the representation of long-tail sparse items.
- **LightGCN** [16] effectively reduces the difficulty of model training by simplifying structure and reducing model complexity while ensuring the effect of graph convolutional network
- **MMGCN** [34] proposes a multimodal graph convolutional network based on the idea of graph neural network messaging, which can generate specific modal representations of users and microvideos to better capture user preferences
- **GRCN** [33] removes the noise information from the interaction graph and applies the graph convolution layer to the refined graph to extract the information signal of user preference.
- **LATTICE** [43] performs graph convolution of the learned potential graphs and explicitly injects higher-order affinities into the item representation.
- **HUIGN** [32] learns multiple levels of user intent from the interaction patterns of items in order to obtain high quality representations of users and items and to further improve recommendation performance.
- **SLMRec** [29] improves the self-supervised learning recommendation effect by designing three data augmentation methods such as feature dropout and feature masking.
- **MICRO** [44] designs a new modal awareness structure learning module and performs graphic convolution to explicitly inject item affinity into the modal awareness item representation.

---

[1] http://jmcauley.ucsd.edu/data/amazon/

- **HCGCN** [26] designs high-order graph convolutions inside user-item and item-item clusters to capture various user behavior patterns.
- **MMGCL** [39] designs two multimodal enhancement techniques to construct multiple views of nodes. In order to ensure the effective contribution of each mode, an effective negative sampling strategy is proposed by pertub one of the modes.

*5.1.3 Evaluation Metrics.* Following related work [26], we use three widely used evaluation indicators, including Recall, Precision and Normalized Discounted Cumulative Gain (NDCG). NDCG is an effective metric to measure ranking tasks. Each metric is calculated based on the top 20 results. The reported results are calculated based on the average of all test users.

*5.1.4 Parameter Protocols.* The dimension of the hidden vector is selected from [16, 32, 64, 128]. Considering the training time and convergence speed, the learning rate is adjusted from [0.00001, 0.00005, 0.0001, 0.0005]. The hyperparameters $\mu, \theta$ and $\gamma$ are searched in [0.0001, 0.001, 0.01, 0.1, 1]. Temperature coefficient $\tau$ is selected from [0.005, 0.05, 0.5, 5, 50]. In addition, the number of intentions is searched from [5, 10, 20, 50, 100].

## 5.2 Overall Performance (RQ1)

We conducted a comprehensive experiment on three amazon data sets and compared our proposed MMIL model with other baseline. The experimental results are shown in Table 2. According to the observation, we can conclude:

- The MMIL model outperforms other state-of-the-art methods in three data sets, and has significant improvement in all evaluation metrics. In particular, compared with VBPR, which simply uses modal information as side-info model, GRCN and MMGCL models which use graph convolution and contrast learning methods have significantly higher effect. Due to the introduction of a better generalization of intention representation method, compared with noise interference representation learning, the multi-intention modeling method can learn user preferences more accurately, making MMIL model more effective than other methods.
- Compared with NCL, MMGCL and other multi-modal recommendation methods based on contrast learning, MMIL has significantly improved the effect. Since most of the previous comparative learning models are based on direct learning of feature representation by means of random sampling, the models are easily affected by noise and mutually exclusive information in the original data. However, MMIL constructs a fine representation of intent from a seemingly coarse-grained perspective, effectively avoiding the interference of noise and irrelevant information. Moreover, the contrast between multiple intentions also improves the accuracy of intention representation.
- MMIL also represents a significant improvement in the three data sets compared to the HUIGN model, which was also built with the intent to make multimodal recommendations. HUIGN thinks that co-item reflects the fine-grained intentions of users, and summarizes the fine-grained intentions

as coarse strength intentions. On the contrary, MMIL explicitly defines intent representation and optimizes the model to learn accurate intent representation through multiple reliable constraint functions. As a result, MMIL can provide more accurate multi-intent recommendations with significantly less computational complexity than HUIGN.

## 5.3 Ablation Study (RQ2)

In order to explore the influence of the proposed modules on the model effect, we consider conducting ablation experiments on three data sets. We conducted the following ablation experiments, and the experimental results are shown in Table 3.

- We removed the intentional learning (IL) module from the model and only retained the feature coding and model prediction parts, denoted as *w/o IL*. We can see that by removing the intention-dependent modules completely, the effect is significantly reduced, similar to the effect of VBPR, which simply uses modal information. It can be seen that modeling intentions is necessary and effective.
- We removed the modal awareness (MA) module from the model and directly use modal independent implicit intention representation, denoted as *w/o MA*. According to the experimental results, the effect of the model decreases after the modal awareness module is removed. This shows that user intentions are strongly related to multimodal information and need to be modeled effectively.
- We droped the intent orthogonal constraint (OC) function from the optimization objective, denoted as *w/o OC*. Without the intention orthogonal constraint, multiple intents are likely to overlap or be similar, which leads to inaccurate intent modeling. The experimental results show that removing intention constraint will damage the effect of model.
- We droped multi-intention contrast (IC) loss from the optimization objective, denoted as *w/o IC*. After the intention contrast loss is removed, the effect of the model decreases on all data sets, indicating the validity of intention contrast. However, in some data sets, the effect changes are not obvious, which indicates that different data sets have different effects of contrast learning due to different data richness.

## 5.4 Parameter Analysis (RQ3)

There are some important parameters in our model. In order to explore the impact of these parameters on the model effect, we conducted sensitivity analysis on the following key parameters.

**Impact of $\mu$ and $\theta$.** Since the parameters $\mu$ and determine the loss weight of intentional representation learning, in order to explore the influence of these two weights on the model effect, we set the values of $\mu$ and $\theta$ in [0.0001, 0.001, 0.01, 0.1, 1]. As shown in Fig. 2, with the increase of parameter values, the performance of the model shows fluctuating changes, without a continuous trend of getting better or worse. In general, the bigger value ensures the effectiveness of intent representation learning, which makes the performance better.

**Impact of $\gamma$ and $\tau$.** Since we introduce multi-intention contrast learning, the parameters $\gamma$ and $\tau$ determine the degree of influence. We let the parameter $\gamma$ adjust the value in [0.0001, 0.001, 0.01, 0.1,

**Table 2: Overall performance comparison of all models on three data sets.**

| Model | Amazon-Clothing | | | Amazon-Sports | | | Amazon-Baby | | |
|---|---|---|---|---|---|---|---|---|---|
| | Recall@20 | Percision@20 | NDCG@20 | Recall@20 | Percision@20 | NDCG@20 | Recall@20 | Percision@20 | NDCG@20 |
| VBPR | 0.0481 | 0.0023 | 0.0205 | 0.0582 | 0.0031 | 0.0265 | 0.0486 | 0.0026 | 0.0213 |
| LightGCN | 0.0470 | 0.0024 | 0.0215 | 0.0782 | 0.0042 | 0.0369 | 0.0698 | 0.0037 | 0.0319 |
| MMGCN | 0.0501 | 0.0024 | 0.0221 | 0.0638 | 0.0034 | 0.0279 | 0.064 | 0.0032 | 0.0284 |
| GRCN | 0.0631 | 0.0032 | 0.0276 | 0.0833 | 0.0044 | 0.0377 | 0.0754 | 0.0040 | 0.0336 |
| MMGCL | 0.0693 | 0.0036 | 0.0307 | 0.0875 | 0.0046 | 0.0409 | 0.0758 | 0.0041 | 0.0331 |
| HUIGN | 0.0735 | 0.0043 | 0.0313 | 0.0865 | 0.0047 | 0.0412 | 0.0761 | 0.0042 | 0.0336 |
| SLMRec | 0.0724 | 0.0041 | 0.0325 | 0.0829 | 0.0043 | 0.0376 | 0.0765 | 0.0043 | 0.0325 |
| LATTICE | 0.0770 | 0.0039 | 0.0316 | 0.0915 | 0.0048 | 0.0424 | 0.0829 | 0.0044 | 0.0368 |
| MICRO | 0.0782 | 0.0040 | 0.0351 | 0.0968 | 0.0051 | 0.0445 | 0.0865 | 0.0045 | 0.0389 |
| HCGCN | 0.0810 | 0.0041 | 0.0370 | 0.1032 | 0.0055 | 0.0478 | 0.0922 | 0.0048 | 0.0415 |
| MMIL | **0.0853** | **0.0043** | **0.0392** | **0.1096** | **0.0059** | **0.0513** | **0.0991** | **0.0052** | **0.0452** |
| Improvement | 5.25% | 4.78% | 5.81% | 6.16% | 6.77% | 7.25% | 7.42% | 8.39% | 8.92% |

**Table 3: Ablation experimental results.**

| Model | Clothing | | Sports | | Baby | |
|---|---|---|---|---|---|---|
| | R@20 | N@20 | R@20 | N@20 | R@20 | N@20 |
| MMIL | **0.0853** | **0.0392** | **0.1096** | **0.0513** | **0.0991** | **0.0452** |
| w/o-IL | 0.0517 | 0.0233 | 0.0625 | 0.0374 | 0.0638 | 0.0306 |
| w/o-MA | 0.0784 | 0.0365 | 0.0977 | 0.0452 | 0.869 | 0.0394 |
| w/o-OC | 0.0828 | 0.0384 | 0.1070 | 0.0496 | 0.0962 | 0.0435 |
| w/o-IC | 0.0813 | 0.0366 | 0.0994 | 0.0481 | 0.0945 | 0.0427 |

1] and $\tau$ adjust the value in [0.005, 0.05, 0.5, 5, 50]. As shown in Fig. 3, as the temperature coefficient increases gradually, the model effect becomes better first and then worse. As the temperature coefficient determines the model's attention to difficult negative samples, appropriate values should be selected for different data sets. In addition, the relatively small $\gamma$ makes the model learn well.

**Impact of embedding dimension $d$.** In order to explore the influence of the implicit vector dimension in the model on the model effect, we adjusted the value of $d$ from [16, 32, 64, 128, 256, 512]. As shown in Fig. 4, As the embedding dimension increases, the performance of the model on all datasets shows a gradual improvement trend. The larger the dimension of implicit vector, the stronger its representation ability, so the effect will be gradually improved. Due to the limited learning ability of the model, the improvement of its effect gradually decreases.

## 6 CONCLUSION

In this paper, we propose a multimodal-aware multi-intention learning method for recommendation. MMIL accurately establishes the relationship between intention and predicted target, and improves recommendation effectiveness through effective intention representation. Specifically, MMIL consists of intention representation module, modal awareness module and intention contrast module. The intentionality representation module introduces implicit intentionality representation and improves intentionality representation learning through intentionality optimization. The modal awareness module models the difference of user's intention distribution among different modes and makes effective use of multi-modal

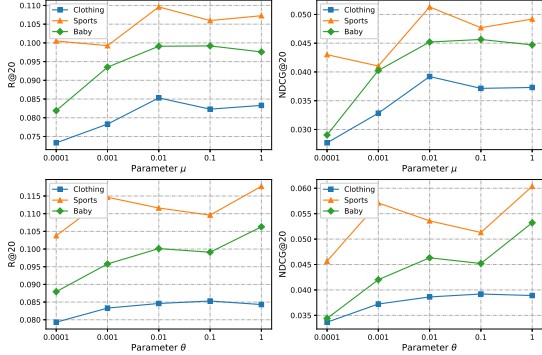

**Figure 2: Effect study of parameters $\mu$ and $\theta$**

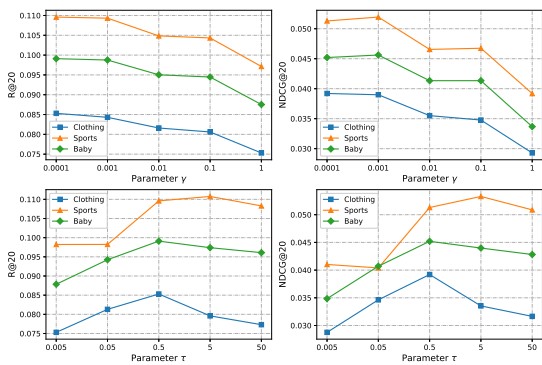

**Figure 3: Effect study of parameters $\gamma$ and $\tau$**

information. Intention contrast module further improves the effect of intention learning through self-supervised learning.

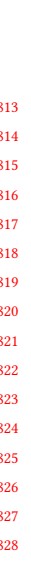

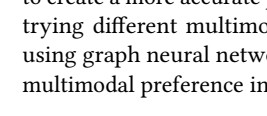

**Figure 4: Effect study of embedding dimension $d$**

In the future, we plan to carry out further research from several directions. First, we consider further exploring the relationship between multimodal information and user intent. We want to try to model the user's multimodal shared intent and modal specific intent to create a more accurate portrait of the user. Secondly, we consider trying different multimodal data utilization methods, including using graph neural network and knowledge graph to mine users' multimodal preference information.

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
