# OpenReview forum: "Multimodal-aware Multi-intention Learning for Recommendation"
_acmmm.org/ACMMM/2024/Conference — MM2024 Poster_

### Official Review · Reviewer_ymqt · 2024-05-20

**Rating:** 4
**Confidence:** 3

**Summary:**

The paper introduces a novel Multimodal-aware Multi-intention Learning (MMIL) method tailored for recommendation systems. It addresses the challenge of effectively utilizing multimodal data in e-commerce and short video platforms to enhance user engagement through personalized recommendations. The authors propose a framework that establishes the relationship between user intentions and recommendation objectives, aiming to avoid intention overfitting and learn accurate multiple intent representations.

**Strengths:**

1. The paper is well-written, with clear logic and motivation.
2. The paper provides extensive and detailed comparative results, demonstrating the effectiveness of the proposed model in multimodal recommendation.
3. The paper includes thorough ablation and hyperparameter experiments, effectively showcasing the impact of the newly designed modules/loss on the overall model.

**Limitations:**

1. The model's loss function design incorporates many sub-losses, requiring cumbersome hyperparameter tuning to find the optimal parameter combination, which could affect its application on real datasets. Additionally, Figure 2 shows that the model is heavily dependent on certain parameter coefficients, and no clear pattern is observed (\theta's performance initially rises, then falls, and then rises again with increasing values).
2. The introduction of the dataset is too simplistic and not self-contained.
3. Sections 5.3 and 5.4 do not specify the datasets used in the experiments. It is recommended to conduct ablation and hyperparameter tuning on multiple datasets.

**Suitability:**

3

---

### Official Review · Reviewer_FLdh · 2024-05-24

**Rating:** 2
**Confidence:** 3

**Summary:**

The paper proposes a Multimodal-aware Multi-intention Learning method (MMIL) for recommendation systems. It aims to improve recommendations by leveraging multimodal data and modeling latent user intentions.

**Strengths:**

The proposed method outperforms existing works in common benchmark datasets.

**Limitations:**

1. The method description in Section 4 is hard to read and follow. Some concept descriptions and illustrations are not clear. What is "I->Z" in Fig.1?
2. The proposed method's detailed network structure, including the backbone network usage and implementation details of intermediate layers (f, g), output layers, etc., is not illustrated.

**Suitability:**

2

---

### Official Review · Reviewer_8aXL · 2024-05-25

**Rating:** 4
**Confidence:** 2

**Summary:**

This paper investigates a multi-intent recommendation scheme that avoids intent overfitting. Specifically, they propose a multimodal-aware multi-intention learning method (MMIL) for recommendation. Experiments showcase the effectiveness of the proposed method.

**Strengths:**

1) The designed method is simple yet effective, and relevant details are provided as well.
2) Adequate experiments are conducted to verify the effectiveness of the solution, comparing various baselines on public datasets and analyzing the parameters.

**Limitations:**

1) The definition and effect of interactions are not discussed, and different interactions, such as viewing, clicking, and adding to cart, are likely to have different impacts on intent modeling.
2) The contribution of contrastive learning may be minimal, as can be seen from the parameter analysis, where setting a very small hyperparameter for CL loss results in better model performance.

**Suitability:**

2

---

### Meta-Review · Area_Chair_BhZz · 2024-06-30

**Recommendation:** Accept (Poster)
**Confidence:** 5

**Metareview:**

Based on the reviewers' suggestions, this work is well-written and has significant value. The authors have responded to the comments, and although the reviewers did not discuss these responses, I believe the authors' replies adequately address the issues raised. Therefore, I recommend accepting this paper.